# The effects of co-designed physical activity interventions in older adults: A systematic review and meta-analysis

Amanda Zacharuk[1], Alison Ferguson[1], Chelsea Komar[1], Nicole Bentley[1], Alexandra Dempsey[1], Michelle Louwagie[1], Sachi O'Hoski[1], Cassandra D'Amore[1], Marla Beauchamp[1,2,3]*

1 School of Rehabilitation Science, McMaster University, Hamilton, ON, Canada, 2 Research Institute at St Joseph's Healthcare, Hamilton, ON, Canada, 3 Department of Medicine, McMaster University, Hamilton, ON, Canada

* beaucm1@mcmaster.ca

**Data Availability Statement:** All relevant data are within the manuscript and its Supporting Information files.

## Abstract

### Background

Physical activity (PA) declines with age despite the knowledge that physical inactivity is a leading cause of disease, death, and disability worldwide. To better tailor PA interventions to older adults, researchers are turning to the collaborative principles of co-design. The purpose of this systematic review was to compare the effectiveness of co-designed PA interventions and standard care for increasing PA and other health outcomes (i.e., physical function, quality of life, mental health, functional independence, attendance and attrition rates) in older adults.

### Methods

A search was conducted in MEDLINE, AgeLine, CINAHL, Embase, and SPORTDiscus. Records were screened by independent pairs of reviewers. Primary research studies conducted among community-dwelling older adults (age 60+) comparing co-designed PA interventions to standard care were considered for inclusion. Controls included wait-list control, usual care, sham interventions, PA interventions without the use of co-design, and no intervention. A random effects meta-analysis was conducted, and the standardized mean difference (SMD) was used to report effect estimates. Quality of evidence was rated using GRADE.

### Results

Of 16,191 studies screened, eight (N = 16,733) were included in this review. Most studies reported results favouring the effect of co-design on physical activity; however, only two studies (N = 433) could be pooled for meta-analysis resulting in a SMD of 0.28, (95% CI = -0.13 to 0.69; p = 0.19; $I^2$ = 56%) immediately post-intervention. The GRADE quality of evidence was very low. The quantitative analysis of three studies reported improved physical function.

**Funding:** The authors received no specific funding for this work.

**Competing interests:** The authors have read the journal's policy and have the following competing interests: MB is supported by a Tier 2 Canada Research Chair in Mobility, Aging and Chronic Disease award (950-233142) as Canada Research Chair (https://www.chairs-chaires.gc.ca/) outside of the submitted work. This does not alter our adherence to PLOS ONE policies on sharing data and materials.

## Conclusion

This review did not demonstrate that co-designed PA interventions are more effective than standard care for increasing PA in older adults; however, evidence was limited and of very low quality. Further well-designed trials are warranted to better understand the impacts of co-designed PA interventions and how to best implement them into practice.

## Trial registration

PROSPERO registration number:
CRD42022314217.

## Introduction

Physical activity (PA) is defined as any bodily movement produced by skeletal muscles that results in energy expenditure [1]. Physical inactivity is responsible for up to 10% of major non-communicable diseases, as well as 9% of premature mortality, making it the fourth leading cause of death globally [2,3]. The cost of physical inactivity to healthcare systems around the world is estimated to exceed INT$50 billion annually, in addition to the indirect effects of the associated disability-adjusted life years and productivity losses [4]. Physical inactivity affects all demographics, but it is especially prevalent in older adults [3,5].

The World Health Organization (WHO) recommends that older adults, including those with disability and chronic conditions, participate in multicomponent PA programs [6]. The weekly recommendations include 150 to 300 minutes of moderate intensity aerobic exercise (or 75 to 150 minutes of vigorous intensity exercise), three moderate intensity strength training sessions, and three days of functional balance training [6]. Achieving age-appropriate levels of PA has been shown to increase physical and mental capacities, improve social outcomes, maintain functional ability, and reduce the risk of disease in older adults [7,8]. Despite the evidence supporting the benefits of PA, the WHO estimates that more than 20% of adults over the age of 60 do not meet the minimum PA guidelines [9]. This number increases to approximately one third of adults between the ages of 70 and 79 and to one half of people over the age of 80 who are insufficiently active [7,9].

It is imperative that we establish effective methods for increasing PA among older adults to improve the health of our aging population and reduce the associated strain on the healthcare system. To provide more appropriate and targeted PA interventions, there is growing interest in including the public in intervention development using co-design [10–12]. Co-design is defined as "a user-centered approach involving collaboration between researchers, end-users, and other relevant stakeholders who are actively engaged throughout a process of iteration and continuous reflection to create an intervention tailored to the specific needs of the target population" [13 p3]. The purpose of co-design is to incorporate the knowledge and experiences of the intended participants into the development of a more effective and sustainable intervention [14]. In doing so, service users feel empowered and valued, researchers develop greater insight into the needs of the target population, and the community gains new skills and experiences [14].

Despite its potential promise, to our knowledge, there are no systematic reviews investigating the effects of co-designed PA interventions on improving PA in older adults [12]. Therefore, the objective of this review was to determine the effects of co-designed PA interventions

on 1) PA levels and 2) other health outcomes including physical function, quality of life, mental health, functional independence, attendance and attrition rates in older adults compared to standard care.

## Methods

The reporting of this systematic review followed the Preferred Reporting Items of Systematic Reviews and Meta-Analyses (PRISMA) guidelines [15,16]. The protocol was registered in PROSPERO (CRD42022314217).

### Search strategy

The search strategy was developed in consultation with our senior investigators (SO, CD, MB) and Health Research Impact Librarian at the McMaster University Health Sciences Library. A search was conducted from inception to February 28th, 2022, in the following electronic databases: MEDLINE (Ovid, 1946 to Present), AgeLine (EBSCOhost, 1978 to Present), CINAHL (EBSCOhost, 1981 to Present), Embase (1974 to Present), and SPORTDiscus (EBSCOhost, 1830 to Present). The primary key terms used to develop the search strategy included physical activity, co-design, and older adults. The full search strategy for each database is included in S1 File. We also conducted a hand search of the reference lists of relevant reviews and studies satisfying the inclusion criteria.

### Inclusion and exclusion criteria

Any primary research studies conducted among community-dwelling older adults that compared co-designed PA interventions to controls were considered for inclusion. In this review, we defined older adults as 60 years of age and older in accordance with the WHO definition [7]. Individuals with any chronic condition were also included to be representative of the older adult population PA had to include bodily movement resulting in energy expenditure. Co-designed PA interventions of any duration, frequency, and intensity that were delivered by any regulated healthcare professional, recreational therapist, or personal trainer that targeted PA levels were included. Co-designed PA interventions must have included at least one older adult in the program development. Comparator groups included wait-list control, usual care, sham interventions, PA interventions without the use of co-design, and no intervention. We chose to look at broad comparators due to the variety of the included populations and the potential for variations in standard care between them. Studies that either compared two groups or used pre-post study designs were included. If pre-intervention data were not presented for pre-post designs, then they were not included. Research poster abstracts, conference abstracts, unpublished or grey literature, and non-English studies were excluded, along with studies that included institutionalized older adults. The primary outcome was PA, based on either self-report or direct measures of enacted performance (i.e., wearables). The secondary outcomes included physical function, quality of life, mental health, functional independence, attendance and attrition rates.

### Study selection

Following the removal of duplicates, the identified studies were imported into the Covidence systematic review software (Veritas Health Innovation, Melbourne, Australia). Title and abstract screening were piloted in consultation with the senior investigators, using 25 abstracts per reviewer. Two independent reviewers screened each remaining abstract for inclusion. The author team then piloted full text screening with 10 full texts before two independent reviewers

screened each remaining article. During the screening process, disagreements were resolved through discussion or through consultation with a third reviewer.

### Data extraction and synthesis

Data extraction and risk of bias assessment were piloted by the review team using two of the included studies. Two independent reviewers performed data extraction and risk of bias assessment for each of the remaining included studies. Data surrounding the methods, population, intervention, control, and outcomes were collected. In the case of any missing data, study author(s) were contacted via email.

The Cochrane Risk of Bias-2 tool was used to assess the risk of bias for each PA outcome within the included randomized control trials (RCTs) and the ROBINS-I tool was used for non-RCTs [17,18]. The Robvis software program (McGuinness, LA, Higgin, JPT) was used to output the risk of bias ratings. Following extraction, data from sufficiently homogenous studies were entered into RevMan 5.4 (Copenhagen: The Nordic Cochrane Centre, Cochrane) for meta-analysis.

Due to the heterogeneity that is common in rehabilitation studies, we planned to use a random-effects approach for any meta-analyses and the standardized mean difference (SMD) to report effect estimates when outcome measures differed. Regarding effect size interpretation, we considered a SMD of 0.2 to be a small effect, 0.5 to be a moderate effect, and 0.8 to be a large effect [19]. Statistical heterogeneity was measured using the $I^2$ statistic. We considered a value of 0–30% minimal heterogeneity; 31–50% moderate heterogeneity; 51–75% substantial heterogeneity and 76–100% considerable heterogeneity [20]. We planned to construct a funnel plot to determine the presence of publication bias if the recommended minimum of 10 studies were included in the meta-analysis [20], and to complete subgroup analyses if there were at least three appropriate studies. We planned to conduct subgroup analyses based on short-term (less than 6-months) versus long-term (greater than 6-months) intervention duration and self-report versus direct measures of PA. Additional outcomes that could not be synthesized quantitatively were included in a systematic narrative synthesis. The quality of the evidence for the primary studies was assessed using the Grading of Recommendations Assessment, Development and Evaluation (GRADE) [21].

## Results

The search yielded 23,954 citations with an additional three studies gathered through hand searching reference lists (Fig 1). After the removal of duplicates, we screened a total of 16,191 abstracts. A total of 158 full texts were assessed for eligibility, of which eight studies were deemed appropriate for inclusion [22–29]. Reasons for exclusion are reported in the PRISMA flow diagram (Fig 1) and the table of excluded studies (S1 Table).

### Study characteristics

Of the eight included studies, four were conducted in the United States [23–25,28] and one each in the Netherlands [22], China [26], Thailand [27], and Japan [29]. Four of the included studies were RCTs [23–26], three were pretest-posttest design [22,27,28], and one was a cluster non-RCT [29]. The sample sizes ranged from 7 to 15,500 participants [22–29]. Year of publication ranged from 2003 to 2021 [22–29]. The mean age of participants ranged from 64.2 to 80.0 years [22–29]. Participants in three studies were from specific population groups (low-vision [22], hypertension [26], fall risk [24]) and the other five studies were general populations of older adults [23,25,27–29].

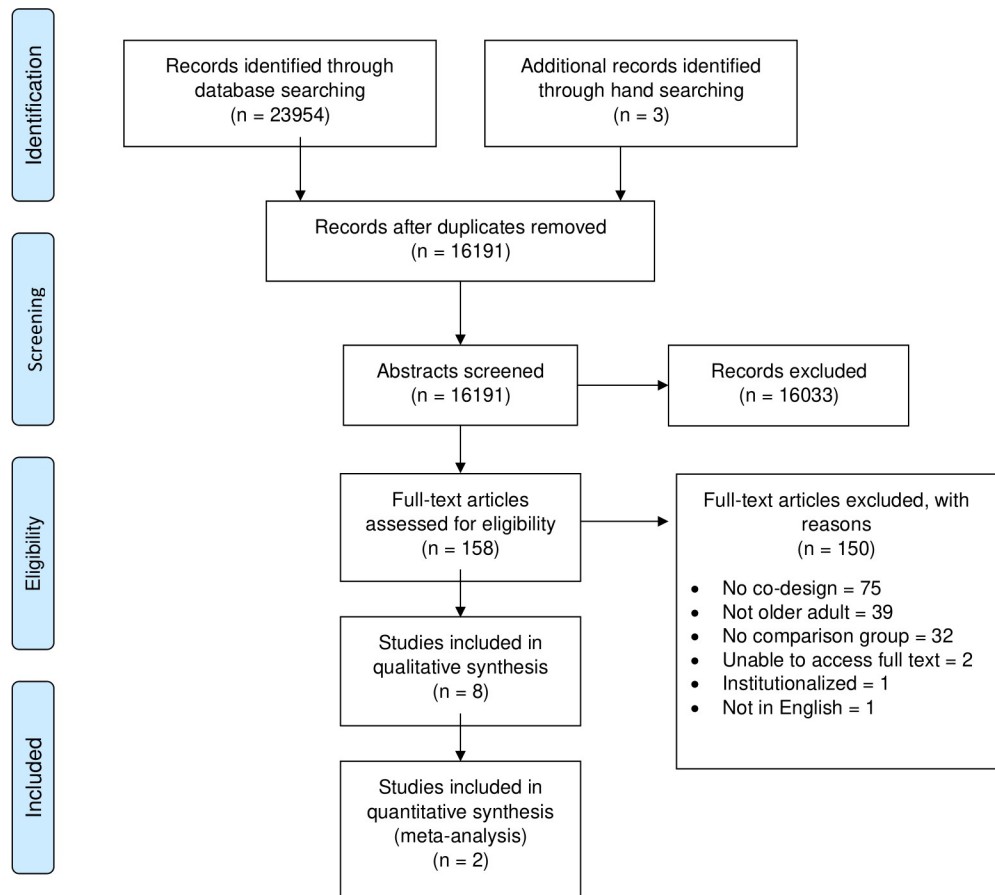

**Fig 1. PRISMA flowchart diagram.** The PRISMA flowchart diagram details the search and study selection process.

The duration of the co-designed interventions ranged from 6 weeks to 2 years and consisted of either specific exercise classes, education, or both [22–29]. Specifically, two interventions focused on tai chi exercise [24,28], two interventions consisted of PA counseling [25,26], one focused on fall prevention education and exercise [27], one focused on frailty and dietary education and exercise [29], and the remaining two studies consisted of general exercise [22,23]. Primary outcome measures varied across studies and included participation [22], physical activity [25,28], function, disability and mobility [23], physical function [24], heart attack and stroke incidence [26], number of falls [27], and frailty [29].

In all eight included studies, co-design was used primarily through participants, or representatives of the participants, acting as stakeholders in collaboration with other health professionals (e.g., occupational therapists, social workers, physiotherapists, physicians, nurses), public health experts, or both, to co-develop the intervention [22–29]. To gain input on the intervention designs, two studies used focus groups [22,28], one study used feedback from a pilot study [29], and another used both focus groups and feedback from a pilot study at the outset of the study [23]. Four studies also included collaboration with stakeholders over the course of the intervention to further inform program development [24–27]. Characteristics of included studies are summarized in Table 1 and further described in S2 Table.

**Table 1. Characteristics of the included studies.**

| Author (Year), Country, [Reference] | Sample Size | % Female | Mean Age | Study Design | Population | PA Intervention[a] | Co-Design Component[b] | Standard Care[c] | Primary Outcomes |
|---|---|---|---|---|---|---|---|---|---|
| Alma (2012), Netherlands, [22] | N = 29 | 69 | 73 | Pretest-Posttest study | Low vision | 20-week exercise/ education program (1/week), 120-minute sessions | Participants and health professionals in focus groups to develop program manual | Pre-intervention data | Participation (USER-P) |
| Brach (2017), USA, [23] | N = 424 (IG: n = 201, SC: n = 223) | 84 | 80 | Cluster RCT | General | 12-week exercise program (2/week), 50-minute sessions | Stakeholders in pilot studies and focus groups to develop intervention. Community Advisory Board members executed exercise intervention | Group exercise program currently conducted at facilities | Function and Disability (LLFDI) Mobility (6MWD, gait speed) |
| Chewning (2019), USA, [24] | N = 242 (IG: n = 123, SC: n = 119) | 84 | 74 | RCT | Falls risk | 6-week exercise/ education program (2/week), 90-minute sessions | Community site coordinators, older adults, and course instructors informed recruitment and course creation | Wait list (received same intervention after study) | Physical Function (30 second chair stand, TUG, 4 stage balance test) |
| Estabrooks (2005), USA, [25] | N = 40 (IG: n = 23, SC: n = 17) | 78 | 77 | RCT | Low income | 12-week exercise group counseling/ goal setting program (1/week), 45-minute sessions | Program administrators and participants collaborated on study design and intervention development | 6 education presentations, 45-minute sessions (2/week) | PA (CHAMPS Physical Activity Questionnaire for Older Adults) |
| Gong (2015), China, [26] | N = 450 (IG: n = 232, SC: n = 218) | 58 | 64 | Longitudinal cluster RCT | Hypertensive | 6-week exercise education/ counseling program (1/week), 10–60-minute sessions, with 2 sessions at 3 months | Public health experts, health professionals, participants, and participants' families participated in program development | Periodic monitoring of health, medication, psychological counseling | Heart attack and stroke (self report of clinical diagnosis) |
| Kittipimpanon (2012), Thailand, [27] | N = 41 | 79 | 73 | Pretest-Posttest study | General | 12-week exercise/ education program (2/week), 45-minute sessions, with 2 sessions within a year | Public health experts and elders involved in situational analysis, fall prevention workshop, and program development | Pre-intervention data | Falls (Fall incidence rate) |
| Perry (2011), USA, [28] | N = 7 | 86 | 70 | Pretest-Posttest study | General | 8-week exercise program (1/week), 60-minute sessions | Older adults and parents of youth involved in survey and focus group for intergenerational PA program | Pre-intervention data | PA (7 DPAR Scale) |
| Sieno (2021), Japan, [29] | N = 15,500 (IG: n = 8000, SC: n = 7500) | 63 | 74 | Cluster non-RCT | General | 2-year education and exercise program | Various professionals including residents, welfare professionals, exercise instructors, "community organizations", companies, research institutions, and government employees discussed and developed the intervention based on a baseline survey and community consultations. | Usual health practices | Frailty status (CL15) |

a The PA intervention utilized.

b The specific aspects of the co-design process used to create the PA intervention.

c The standard care comparator used.

Abbreviations: USA = United States of America, PA = physical activity, RCT = randomized controlled trial, CB = community-based, HB = home-based, IG = intervention group, SC = standard care, USER-P = Utrecht Scale for Evaluation of Rehabilitation Participation, CHAMPS = Community Healthy Activities Model Program for Seniors, IPA = impact on participation and autonomy, L-LFDI = Late-Life Function and Disability Instrument, 6MWD = 6 minute walk distance, ABC = Activities-specific Balance Confidence, TUG = Timed up and go, 7 DPAR = 7-Day Physical Activity Recall, CL15 = Check-List 15.

## Risk of bias

Risk of bias findings for the included studies are summarized in S1 Fig. All RCTs were judged to be high risk of bias. The randomization and allocation concealment processes increased risk of bias in all four RCTs, three of which were rated as high risk [23–26]. Two of the four RCTs deviated from the intended intervention [23,26]. All RCTs included some form of self-report measure, which increased the risk of bias since assessors could not be blinded [23–26]. The risk of bias due to selective outcome reporting was judged to be moderate in one study as we were unable to locate the protocol to establish consistency with predefined methods [24].

Among the non-RCTs, one study was found to be at critical risk [28], two at high risk [22,29], and one at moderate risk of bias [27]. Two of the five studies were rated moderate risk of bias due to confounding factors [22,29]. All four trials reported an adequate participant selection process and classification of interventions. Regarding risk of bias due to deviations from the intended intervention, one study was judged as high risk [22]. The risk of bias due to missing data was critical for one study because the attrition rate was greater than 20% [28]. Three non-RCT studies used self-report measures [22,28,29] and one study did not blind outcome assessors [27] which increased the risk of bias in measurement outcomes. Risk of bias in selection of reported results in one study was unclear [28], but low risk in the other three studies [22,27,29].

## Primary outcome

The primary outcomes reported in the included studies are provided in Table 1. Our primary outcome of interest was PA. Meta-analyses of two studies [22,26] were conducted for effects on PA immediately post intervention and at 6-months post intervention. Only two studies could be pooled for meta-analysis due to missing raw data in the other six included studies. We attempted to contact the corresponding authors of four studies [25,27–29], but were unable to obtain the required data. Results from the other six studies were included in the qualitative synthesis. Overall, five studies reported an increase in PA favouring co-design while one study did not.

The random effects meta-analysis for effects of co-designed interventions on PA immediately post intervention including 433 participants was not statistically significant (SMD = 0.28; 95% CI = -0.13 to 0.69; p = 0.19) (Fig 2). There was substantial inconsistency ($I^2$ = 56%; Chi-squared = 2.29, degrees of freedom [df] = 1, p = 0.13). At 6-months follow-up, the random effects meta-analysis including 407 participants was also not statistically significant (SMD = 0.20; 95% CI = -0.39 to 0.79; p = 0.51) (Fig 3). The pooled estimate had considerable inconsistency ($I^2$ = 76%) and significant statistical heterogeneity (Chi-squared = 4.26, df = 1, p = 0.04). No subgroup or sensitivity analyses were conducted due to the limited number of studies.

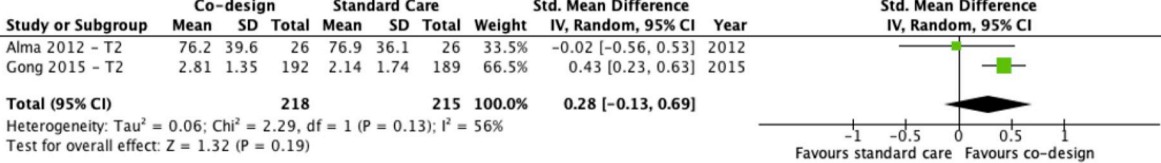

**Fig 2. Forest plot of the effect of co-designed physical activity interventions on self-reported levels of physical activity in older adults immediately post intervention.** Standardized mean difference, 95% confidence interval and results of tests for heterogeneity are presented.

| Study or Subgroup | Co-design Mean | SD | Total | Standard Care Mean | SD | Total | Weight | Std. Mean Difference IV, Random, 95% CI | Year |
|---|---|---|---|---|---|---|---|---|---|
| Alma 2012 – T1 | 71.5 | 30 | 26 | 76.9 | 36.1 | 26 | 41.3% | −0.16 [−0.70, 0.38] | 2012 |
| Gong 2015 – T1 | 3.37 | 1.28 | 183 | 2.66 | 1.81 | 172 | 58.7% | 0.45 [0.24, 0.67] | 2015 |
| **Total (95% CI)** | | | **209** | | | **198** | **100.0%** | **0.20 [−0.39, 0.79]** | |

Heterogeneity: Tau² = 0.14; Chi² = 4.26, df = 1 (P = 0.04); I² = 76%
Test for overall effect: Z = 0.66 (P = 0.51)

**Fig 3. Forest plot of the effect of co-designed physical activity interventions on self-reported levels of physical activity in older adults at 6-months follow-up.** Standardized mean difference, 95% confidence interval and results of tests for heterogeneity are presented.

A pretest-posttest study by Alma et al. (2012) used the Utrecht Scale for Evaluation of Rehabilitation-Participation (USER-P) self-report questionnaire [22]. The USER-P contains an item for the number of times the client had participated in physical activity in the last 4 weeks. Mean scores decreased over time from baseline [mean (SD) = 76.9 (36.1)] to immediately post intervention [mean (SD) = 76.2 (39.6)] and at 6-months follow-up [mean (SD) = 71.5 (30.0)], demonstrating that the co-design PA intervention in Alma et al. (2012) did not have an effect on PA in comparison to the pre-test group [22]. A cluster RCT conducted by Gong et al. (2015) used a self-report five-point rating scale to quantify the average duration of exercise in a typical day [26]. At 3-months, significantly greater levels of PA were achieved by the intervention group [mean (SD) = 2.81 (1.35)] compared to the non-co-design group [mean (SD) = 2.14 (1.74); Cohen's $d$ = 0.53, 95% CI: 0.21, 0.85]. At 6-months post intervention, the intervention group had significantly greater levels of PA [mean (SD) = 3.37 (1.28)] compared to the non-co-design group [mean (SD) = 2.66 (1.81); Cohen's $d$ = 0.45, 95% CI: 0.04, 0.85] [26]. An RCT by Estabrooks et al. (2005) assessed PA using caloric expenditure and the CHAMPS Physical Activity Questionnaire for Older Adults [25]. The intervention group significantly increased their weekly PA compared to the non-co-design group (p<0.05, ES = 0.79). The intervention group increased PA caloric expenditure (from 1,610.9 to 2,676.3 kcal), while the non-co-design group did not (from 1,597.6 to 1,317.1 kcal) [25]. A pretest-posttest study by Perry et al. (2011) assessed PA using the 7-day physical activity recall which is used to recall PA over the last 7 days (including planned and unplanned exercise). Perry et al. (2011) reported that their co-design PA intervention had the potential to increase overall PA when compared to a pre-test group; however, data were only presented in the form of a scatterplot and, therefore, the raw data could not be analyzed [28]. Despite this limitation, the scatterplot displayed a positive trend toward increasing overall PA [28]. The quasi-experimental study by Seino et al. (2021) assessed PA using the percentage of those who exercised more than once per week, and walked over 150 minutes per week as measured by self-report [29]. At baseline, 73.9% of participants in both the control group and intervention group exercised one or more times per week. At 2-year follow-up, these percentages increased to 74.1% for the control group and 75.4% for the intervention group [29]. At baseline, 71.9% and 70.2% of participants in the control group and intervention group respectively, walked 150 minutes or more per week. At 2-year follow-up, this percentage increased to 80.1% for both groups. Although these findings trended towards more positive health behaviours, they were not statistically significant. The subgroup analysis was conducted only for the participants in one district, and a statistically significant change was found in walking minutes after the 2-year follow-up (p = 0.001) [29]. All measures of PA would have included the time spent in the exercise intervention for all studies [22–29].

## GRADE

Using GRADE, the certainty of the evidence from the meta-analyses was judged to be very low. The evidence was downgraded due to high risk of bias, imprecision (due to small sample

sizes and large confidence intervals), and indirectness of the included population (due to specific clinical groups included in these two studies). We were unable to construct a funnel plot because fewer than 10 studies were included. Refer to S2 Fig for the GRADE summary table.

### Secondary outcomes

The secondary outcomes of interest were physical function, mental health, functional independence, attendance and attrition rates. No studies reported on functional independence or quality of life. The qualitative synthesis is reported below for the other secondary outcomes. Meta-analyses could not be conducted for these outcomes as we were unable to extract usable data from these studies or retrieve original data from the study authors.

**Physical function.** Two RCTs [23,24] and one quasi experimental study [27] assessed the effect of a co-designed PA intervention on physical function through performance-based measures. These outcome measures included: the Timed Up and Go (TUG) [24,27], grip strength [27], 360-degree turn [27], 5 Times Sit to Stand [27], grip strength test [27], 30-second Chair Stand [24], 6 Minute Walk Distance (6MWD) [23], and gait speed [23]. The RCT by Chewning et al. (2019) found a significant between-group improvement for the 30-second Chair Stand test, and the RCT by Brach et al. (2017) found a significant between-group improvement in 6MWD and gait speed scores when compared to their respective control groups. Both Kittipimpanon et al. (2012) and Chewning et al. (2019) reported a significant improvement in TUG scores in favour of the co-designed PA interventions. The quasi-experimental study by Kittipimpanon et al. (2012) also reported a significant within-group post intervention improvement for the 5 Times Sit to Stand and 360-degree turn, but no significant change was reported for grip strength between pre and post intervention [27]. The findings from these studies suggest a positive effect on most performance-based measures of physical function in favour of co-designed PA interventions.

One RCT and one quasi experimental study measured physical function through self-report. The RCT by Brach et al. (2017) measured physical function by the Late-Life Function and Disability Instrument and reported no significant difference between groups [23]. The quasi-experimental study by Seino et al. (2021) measured physical function by the motor fitness scale and mobility limitation (self-reported difficulty in walking 0.4km or climbing 10 steps without resting) and reported no significant difference between groups [29].

**Mental health.** Mental health was only reported in one study via the geriatric depression scale and the World Health Organization-Five Well-being Index [29]. Their results did not show any significant effects on mental health outcomes between groups [29].

**Attendance and attrition rates.** Five studies measured attrition rates and program attendance. Gong et al. (2015) had an adherence rate of 82.8% post intervention and 78.9% at 6-months [26]. In Perry et al. (2011), 30% withdrew from the program for reasons not reported [28]. Kittipimpanon et al. (2012) had 28 participants in the pre-test group and after follow-up, suggesting evidence of no attrition [27]. The study by Brach et al. (2017) reported attendance by the number of participants who attended 20 or more classes throughout the program [23]. In the non-co-design group, 95 individuals attended 20 or more classes compared to 76 individuals in the co-design PA group [23]. Chewning et al. (2019) reported attendance based on the mean number of classes attended by experimental group participants, which was 11 out of the 12 total classes [24].

## Discussion

The purpose of this review was to determine the effects of co-designed PA interventions on PA levels and other health outcomes including physical function, quality of life, mental health,

functional independence, attendance and attrition rates in older adults relative to standard care. Based on very low-quality evidence from the meta-analyses of two studies [22,26], this review did not demonstrate an effect of co-designed PA interventions on levels of PA compared to controls. However, the pooled effect size and narrative results for our primary outcome suggest that more research is warranted. Our qualitative synthesis for the secondary outcomes of physical function and quality of life also favoured co-design, although no definitive conclusions could be drawn without meta-analysis.

Although the pooled results were not statistically significant, the SMDs for the effects of co-designed PA interventions both immediately post-intervention (SMD = 0.28) and at 6-months follow-up (SMD = 0.20) suggest that we cannot rule out that the interventions may have had a small positive effect on improving PA. It is important to note that the two studies included in the meta-analyses focused on specific clinical populations (i.e., individuals with visual impairment and individuals with hypertension) which may partly explain the heterogeneity and small effect sizes [22,26]. Grbović and Stanimirov (2020) found that adults with visual impairment report common barriers to PA such as a lack of knowledge and skills, the physical environment, and factors related to the social environment [30]. Additional barriers in this population include the visual impairment itself and other psychological factors [30]. Similarly, Churilla and Ford (2010) found that only 60% of hypertensive adults in the United States met the Department of Health and Human Services recommendations for PA [31]. Although most participants with hypertension did engage in exercise, they were still less active compared to those without hypertension [31]. Thus, improving PA may be particularly challenging for individuals with visual impairment and for individuals with hypertension. Our qualitative synthesis included one study with a low income population [25], one study with a falls risk population [24] and four studies with general populations of older adults [23,27–29]. Co-designed PA interventions improved PA in five out of the six studies included in the qualitative synthesis. Therefore, our findings suggest that further well-designed trials are warranted to investigate the use of co-design for improving PA among the general older adult population.

Data from studies included in the qualitative synthesis suggest that co-designed PA interventions improved performance-based measures (i.e., 30 second Chair Stand test, 6MWD, gait speed, TUG) [23,24,27] and self-report measures (i.e., LLFDI, motor fitness scale, mobility limitations) [23,29] of physical function relative to controls. Both gait speed and the TUG can be used to predict a decline in global health, physical function, and falls in community-dwelling older adults [32]. Poor performance on these outcomes is associated with an increased risk of hospitalization and a decline in quality of life [32]. Co-designed PA interventions also improved performance on the 30-second Chair Stand test and 6MWD [23,24], which are important indicators of lower extremity function, fitness, and frailty [33,34]. Overall, more trials are needed to compare co-designed and non-co-designed PA interventions to further investigate their impact on older adults' physical function.

Regarding program attendance, varied reporting methods (attrition rates vs attendance) and the lack of data from pretest-posttest and waitlist control studies limited our ability to meaningfully synthesize these findings [22,24,27,28]. In a review investigating older adults' exercise adherence, the average rate of adherence to the intervention was 78.2% [35]. Gong et al. (2015) and Chewning et al. (2019) both reported higher attendance rates for the co-designed PA interventions compared to the non-co-design group and wait-list control group [24,26]. Comparatively, one study reported greater attendance in the non-co-design group [23]. In the study by Brach et al. (2017), the control group participated in an exercise program with most of the exercises in sitting, compared to the co-designed intervention group, who performed most exercises in standing. Therefore, these contradictory findings might be explained

by the difference in difficulty level between the two interventions. Further research is required to compare program attendance with co-designed versus non-co-designed PA interventions.

Co-design implementation varied throughout the studies included in this review. Some similarities related to involving participants, or representatives of the participants, as stakeholders in the collaboration and development of the intervention. However, the methods of the co-design processes differed in the members that made up the co-design team, the use of various design components (e.g., pilot groups, focus groups, or a combination of both), and the duration of member involvement in the co-design process. This theme of heterogeneity has also been recognized in other literature, as co-designed PA studies in older adult populations have demonstrated the use of co-design through various means, such as focus groups, workshops, surveys, and interviews, with inconsistent amounts of collaboration with participants between studies [36–38]. These inconsistencies highlight the need for a systematic approach to co-design to allow for better generalizability of co-design principles [12]. The lack of a systematic approach in the included studies in our review increased heterogeneity and decreased the generalizability of our findings. Further evaluation of the relationship between the level of involvement and the specific processes used in co-designed PA interventions could lead to a better understanding of the key co-design components.

## Strengths and limitations

The review was conducted in accordance with PRISMA guidelines and registered in PROSPERO to enhance the quality of reporting and overall rigor. Our comprehensive search strategy was reviewed by a health research librarian and a breadth of studies were obtained through the literature search. The deviations from the registered protocol included the inability to conduct subgroup or sensitivity analyses or a funnel plot due to the limited number of included studies. All deviations were reported and justified above to provide transparency and reduce bias in reporting. To maintain strong methodological quality, we completed a full risk of bias assessment to evaluate the quality of research and employed the GRADE approach to assess the certainty of the evidence.

All eight included studies had an overall high risk of bias which limits our ability to draw conclusions. Our intention had been to exclude studies with an overall high risk of bias from the meta-analysis but, due to the limited number of eligible studies, we had to include them to pool results. Within the included community-dwelling older adults, we did not restrict the population based on pre-existing health conditions in hopes of including a broad range of participants to portray the general older adult population. However, the two studies that could be pooled for meta-analysis included older adults with specific clinical conditions (i.e., low-vision and hypertension). The recruitment settings for the included studies varied (e.g., gym, low-income housing). Non-English literature was not included, resulting in language bias. Despite the associated risk of publication bias, non-published literature was excluded to promote the inclusion of higher quality and peer-reviewed studies.

## Future research and implications

Based on the current body of literature, concrete recommendations cannot be made regarding the implementation of co-designed PA interventions to increase PA in practice. However, since co-design incorporates many aspects of client-centered care [39], co-designed PA interventions may be an asset in the delivery of high-quality rehabilitation and health promotion programs. More rigorous research is required to better understand the impacts of co-designed PA interventions and how to best implement them into practice. Additionally, future studies should explicitly measure changes in total amounts of PA (e.g., pedometers) to increase methodological rigor and allow for blinding of assessors. Additionally, trials with longer follow-up

are needed to investigate the effectiveness of co-designed PA interventions over the longer-term.

## Conclusion

In summary, the meta-analysis did not demonstrate a statistically significant effect of co-designed PA interventions on PA; however, we cannot rule out the possibility of a small positive effect on improving PA. These quantitative findings were based on very low-quality evidence, and results should be interpreted with caution. Our qualitative synthesis did support co-designed PA interventions for improving PA and physical function; however, an overall effect estimate could not be measured. Further trials are needed to better understand the impacts of co-designed PA interventions and how to best implement them into practice.

## Supporting information

**S1 File. Search strategy.**
(DOCX)

**S2 File. PRISMA checklist.**
(DOCX)

**S1 Table. List of excluded studies and reasons for exclusion.**
(DOCX)

**S2 Table. Characteristics of included studies.**
(DOCX)

**S1 Fig. Risk of bias summaries.**
(PDF)

**S2 Fig. GRADE summary.**
(DOCX)

## Acknowledgments

Rachel Couban, Health Research Impact Librarian at the McMaster University Health Sciences Library, for her assistance with developing the search strategy.

## Author Contributions

**Conceptualization:** Amanda Zacharuk, Alison Ferguson, Chelsea Komar, Nicole Bentley, Alexandra Dempsey, Michelle Louwagie, Sachi O'Hoski, Cassandra D'Amore, Marla Beauchamp.

**Data curation:** Amanda Zacharuk, Alison Ferguson, Chelsea Komar, Nicole Bentley, Alexandra Dempsey, Michelle Louwagie, Sachi O'Hoski, Cassandra D'Amore, Marla Beauchamp.

**Formal analysis:** Amanda Zacharuk, Alison Ferguson, Sachi O'Hoski, Cassandra D'Amore, Marla Beauchamp.

**Investigation:** Amanda Zacharuk, Alison Ferguson, Chelsea Komar, Nicole Bentley, Alexandra Dempsey, Michelle Louwagie, Sachi O'Hoski, Cassandra D'Amore.

**Methodology:** Amanda Zacharuk, Alison Ferguson, Chelsea Komar, Nicole Bentley, Alexandra Dempsey, Michelle Louwagie, Sachi O'Hoski, Cassandra D'Amore, Marla Beauchamp.

**Project administration:** Amanda Zacharuk, Alison Ferguson, Chelsea Komar, Nicole Bentley, Alexandra Dempsey, Sachi O'Hoski, Cassandra D'Amore, Marla Beauchamp.

**Resources:** Sachi O'Hoski, Cassandra D'Amore, Marla Beauchamp.

**Supervision:** Sachi O'Hoski, Cassandra D'Amore, Marla Beauchamp.

**Validation:** Sachi O'Hoski, Cassandra D'Amore, Marla Beauchamp.

**Visualization:** Amanda Zacharuk, Alison Ferguson, Chelsea Komar, Nicole Bentley, Alexandra Dempsey, Michelle Louwagie, Sachi O'Hoski, Cassandra D'Amore, Marla Beauchamp.

**Writing – original draft:** Amanda Zacharuk, Alison Ferguson, Chelsea Komar, Nicole Bentley, Alexandra Dempsey, Michelle Louwagie, Sachi O'Hoski, Cassandra D'Amore, Marla Beauchamp.

**Writing – review & editing:** Amanda Zacharuk, Alison Ferguson, Chelsea Komar, Sachi O'Hoski, Cassandra D'Amore, Marla Beauchamp.

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
