## [Decision Letter · Decision Letter 0]

27 Sep 2023

PONE-D-23-07939The effects of co-designed physical activity interventions in older adults: a systematic review and meta-analysisPLOS ONE

Dear Dr. Beauchamp

Thank you for submitting your manuscript to PLOS ONE. After careful consideration, we feel that it has merit but does not fully meet PLOS ONE’s publication criteria as it currently stands. Therefore, we invite you to submit a revised version of the manuscript that addresses the points raised during the review process.

We look forward to receiving your revised manuscript.

Kind regards,

Opeyemi O Babatunde, Ph.D., MPh., B.Physio

Academic Editor

PLOS ONE

Reviewers' comments:

Reviewer's Responses to Questions

**Comments to the Author**

1. Is the manuscript technically sound, and do the data support the conclusions?

Reviewer #1: Yes

Reviewer #2: Yes

2. Has the statistical analysis been performed appropriately and rigorously? 

Reviewer #1: Yes

Reviewer #2: Yes

3. Have the authors made all data underlying the findings in their manuscript fully available?

Reviewer #1: Yes

Reviewer #2: Yes

4. Is the manuscript presented in an intelligible fashion and written in standard English?

Reviewer #1: Yes

Reviewer #2: Yes

5. Review Comments to the Author

Reviewer #1: Thank you for the opportunity to review this paper. In general, this paper was well written, and provided insight to a fairly new area in physical activity research. Please see below for some suggestions:

ABSTRACT

- Background - your aim states that "the purpose of the review was to compare the effectiveness..." but you don't state WHAT you comparing it with.

INTRODUCTION

- Overall very well written. I would just recommend clarifying the aim so that it is reflected in the abstract. I would also add that you are comparing with standard care.

METHODS

- In the inclusion/exclusion criteria, you identify that individuals with chronic conditions were included - could you list the type of conditions that were included? And also the ones that were excluded - with reasons.

RESULTS

- Summary of results were clear

- Table 1 - good description of studies, but I would like clarification on the difference between intervention and program development

- Meta-analysis - I am concerned that meta-analysis was completed on two very different studies. One study was a pre-post test and the other was an RCT. Furthermore, it was unclear from the table and text whether it was appropriate to pool the studies given their different PA outcomes. Could you please outline whether this was appropriate? Please provide justification.

- Primary outcomes p.10 - please report the PA outcome measures reported by the studies in the text or refer to Table 1.

- Qualitative description of review p. 11 - the authors made broad, sweeping claims, such as "co-design did not have effect on PA in this study". Please be more specific. For example, was it co-design or a co-designed PA intervention? What was the control for this? It's important to have this kind of information so that the reader is well-informed about the claims.

This was the same for the secondary outcomes on p. 12. Please provide further information.

DISCUSSION

- Similar to your results, authors need to provide further detail. Modifying the results may help with improvements in the discussion.

Reviewer #2: I believe this is a well conducted research. The title is unambiguous, and the research methods used in achieving the aims and objectives are appropriate. The researchers explained in details their set objectives and aims, measured outcomes before the onset of the study, such that the study will be easy to duplicate. They explained the rationale for each decision they took, and identified sources of their data. The conclusion was quite clear and precise, and they acknowledged the strengths and limitations of the study.

However, I believed the study where participants had only counselling, should be made more explanatory, as not all counselling involve exercises, some may just be participants sharing their thoughts with a therapist.

On the long run, I believe the article should be accepted and published.

6. PLOS authors have the option to publish the peer review history of their article (what does this mean?). If published, this will include your full peer review and any attached files.

Reviewer #1: No

Reviewer #2: No

---

## [Author Response · Author response to Decision Letter 0]

9 Jan 2024

Reviewer #1:

Comment: Thank you for the opportunity to review this paper. In general, this paper was well written, and provided insight to a fairly new area in physical activity research. Please see below for some suggestions:

Response: Thank you for the opportunity to address these helpful comments and to resubmit our work. We look forward to a favorable response regarding our revised manuscript. We have responded to each of your comments below.

Comment: ABSTRACT

- Background - your aim states that "the purpose of the review was to compare the effectiveness..." but you don't state WHAT you comparing it with.

Response: Our aim statement in the abstract describes that we are comparing co-design physical activity interventions to standard care. We revised the statement to remove the word “relative” and relocated the comparator of standard care in the sentence in hopes of clearing any confusion. This revision can be found on Page 2 Lines 18-21: “The purpose of this systematic review was to compare the effectiveness of co-designed PA interventions and standard care for increasing PA and other health outcomes (i.e., physical function, quality of life, mental health, functional independence, attendance and attrition rates) in older adults.”

Comment: INTRODUCTION

- Overall very well written. I would just recommend clarifying the aim so that it is reflected in the abstract. I would also add that you are comparing with standard care.

Response: We revised accordingly. Page 3 Line 87-90: “Therefore, the objective of this review was to determine the effects of co-designed PA interventions on 1) PA levels and 2) other health outcomes including physical function, quality of life, mental health, functional independence, attendance and attrition rates in older adults compared to standard care.”

Comment: METHODS

- In the inclusion/exclusion criteria, you identify that individuals with chronic conditions were included - could you list the type of conditions that were included? And also the ones that were excluded - with reasons.

Response: We wanted to be as inclusive as possible in order to represent the older adult population. For this reason, we did not exclude studies of individuals with specific chronic conditions. In the methods section we outline this but, based on the feedback, we also revised the wording to further clarify on Page 4 Lines 114-115: “Individuals with any chronic conditions were also included to be representative of the older adult population.” A list of the chronic conditions that ended up being included are presented within the study characteristics section of the results on Page 6 Line 193-195: “Participants in three studies were from specific population groups (low-vision [22], hypertension [26], fall risk [24]) and the other five studies were general populations of older adults [23,25,27–29].”

Comment: RESULTS

- Summary of results were clear

Response: Thank you.

Comment: - Table 1 - good description of studies, but I would like clarification on the difference between intervention and program development

Response: On Page 8 Table 1 the heading “Program” was revised to “PA Intervention” to be more clear for the reader. Additionally, descriptions for the PA intervention, co-design component, and standard care titles were added to the bottom of Table 1 to clarify the information provided in each column. We wanted to emphasize these components to lay out our PICO for each study so they could easily be compared. These revisions can be found in the footnote on Page 10 Table 1: 

“a The specific PA intervention utilized.

b The specific aspects of the co-design process used to create the PA intervention.

c The standard care comparator used.”

Comment: - Meta-analysis - I am concerned that meta-analysis was completed on two very different studies. One study was a pre-post test and the other was an RCT. Furthermore, it was unclear from the table and text whether it was appropriate to pool the studies given their different PA outcomes. Could you please outline whether this was appropriate? Please provide justification.

Response: We understand your concern, and this is something we discussed in great detail. We realize there are limitations with pooling results for these two studies; however, we outlined these limitations in our strengths and limitations section. We also used a random effects approach and standardized mean difference (SDM) to account for the heterogeneity for these two studies, as recommended by the Cochrane review guidelines.

We decided to leave the meta-analysis as it is. We feel these two studies are sufficiently homogenous. They both delivered a co-design physical activity intervention designed to promote increased physical activity. Physical activity level in both studies was measured by self-report using a likert scale and both groups were compared to a standard care group (one being a control group and one a pre-test group). Other than the different types of study designs, the one other difference was the population (one involving older adults with hypertension and the other involving visually-impaired older adults), but we did not exclude chronic conditions from our study overall. If this remains an issue, we will be happy to revisit this comment and discuss further.

Comment: - Primary outcomes p.10 - please report the PA outcome measures reported by the studies in the text or refer to Table 1.

Response: We revised accordingly. Page 11 Line 241: “The primary outcomes reported in the included studies are provided in Table 1”.

Comment: Qualitative description of review p. 11 - the authors made broad, sweeping claims, such as "co-design did not have effect on PA in this study". Please be more specific. For example, was it co-design or a co-designed PA intervention? What was the control for this? It's important to have this kind of information so that the reader is well-informed about the claims.

This was the same for the secondary outcomes on p. 12. Please provide further information.

Response: Thank you for your feedback. We reviewed the reporting of our results and made our statements more clear to ensure that we were more explicit in our language. On Page 12 Line 272-275 “Mean scores decreased over time from baseline [mean (SD) = 76.9 (36.1)] to immediately post intervention [mean (SD) = 76.2 (39.6)] and at 6-months follow-up [mean (SD) = 71.5 (30.0)], demonstrating that the co-design PA intervention in Alma et al. (2012) did not have an effect on physical activity in comparison to the pre-test group”. This revised sentence clarifies that we are referring to the results from the specific study by Alma et al. (2012). The study was a pre-test post-test design; therefore, the comparison group was the pre-test group. 

We reviewed the narrative description of the included studies' findings for the various outcomes, and made several amendments to improve clarity and ensure specificity. On Page 12 Line 288-291: “Perry et al. (2011) reported that their co-design PA intervention had the potential to increase overall PA when compared to a pre-test group”. ‘Pre-test group’ was added at the end of the sentence to clarify what control was used. We added a statement so the readers are aware which study design was conducted by Seino et al (2021) on Page 12 Line 292: “The quasi-experimental study by Seino et al. (2021) assessed PA using the percentage of those who exercised more than once per week, and walked over 150 minutes per week as measured by self-report.” A statement about the control groups was added on Page 13 Line 328-331: “The RCT by Chewning et al. (2019) found a significant between-group improvement for the 30-second Chair Stand test, and the RCT by Brach et al. (2017) found a significant between-group improvement in 6MWD and gait speed scores when compared to their respective control groups”. 

Comment: DISCUSSION

- Similar to your results, authors need to provide further detail. Modifying the results may help with improvements in the discussion.

Response: We adjusted our methods and results, as suggested, and we affirmed consistent language in the discussion. We are hesitant to make further changes to the discussion to avoid redundancy. We would be happy to revise if the reviewer can provide further guidance, or if they have more specific suggestions.

Reviewer #2:

Comment: I believe this is a well conducted research. The title is unambiguous, and the research methods used in achieving the aims and objectives are appropriate. The researchers explained in details their set objectives and aims, measured outcomes before the onset of the study, such that the study will be easy to duplicate. They explained the rationale for each decision they took, and identified sources of their data. The conclusion was quite clear and precise, and they acknowledged the strengths and limitations of the study.

Response: Thank you for the opportunity to address these helpful comments and to resubmit our work. We look forward to a favorable response regarding our revised manuscript. We responded to each of your comments below.

Comment: However, I believed the study where participants had only counselling, should be made more explanatory, as not all counselling involve exercises, some may just be participants sharing their thoughts with a therapist.

Response: A statement was added to the inclusion/exclusion criteria to more clearly outline that the PA intervention did not need to be conducting PA but needed to target PA levels on Page 4 Line 115-118: “Co-designed PA interventions of any duration, frequency, and intensity that were delivered by any regulated healthcare professional, recreational therapist, or personal trainer that targeted PA levels were included.” Clarification was added to the brief descriptions on the PA interventions since the counseling interventions did target PA levels on Page 8 Table 1: for Estabrooks (2005) “12-week exercise education/ group counseling/goal setting program (1/week), 45-minute sessions” and for Gong (2015) “6-week exercise education/ counseling program (1/week), 10–60-minute sessions, with 2 sessions at 3 months”. As well, further descriptions of the studies, including the interventions, are available in the supplemental appendix S3. Appendix S3 was revised to clarify the PA intervention for Gong (2015): “IG: Two group lectures, two sessions of one-on-one telephone counseling, and two group meetings focused on addressing PA barriers, improving self-efficacy, increasing self-motivation, and empowering participants to increase their PA levels. After the 3-month follow-up assessment, a two-session booster was arranged starting with telephone-counseling followed by a group meeting. The IG also received the SC.”

Comment: On the long run, I believe the article should be accepted and published.

Response: Thank you.

---

## [Editor Report · Decision Letter 1]

11 Jan 2024

The effects of co-designed physical activity interventions in older adults: a systematic review and meta-analysis

PONE-D-23-07939R1

Dear Dr. Beauchamp,

We’re pleased to inform you that your manuscript has been judged scientifically suitable for publication and will be formally accepted for publication once it meets all outstanding technical requirements.

Kind regards,

Opeyemi O Babatunde, Ph.D., MPh., B.Physio

Academic Editor

PLOS ONE

Additional Editor Comments (optional):

Well done!
---

## [Editor Report · Acceptance letter]

29 Apr 2024

PONE-D-23-07939R1 

PLOS ONE

Dear Dr. Beauchamp, 

I'm pleased to inform you that your manuscript has been deemed suitable for publication in PLOS ONE. Congratulations! Your manuscript is now being handed over to our production team.

Kind regards, 

on behalf of

Dr. Opeyemi O Babatunde 

Academic Editor

PLOS ONE